# On the Convergence and Calibration of Deep Learning with Differential Privacy

## Abstract

In deep learning with differential privacy (DP), the neural network achieves the privacy usually at the cost of slower convergence (and thus lower performance) than its non-private counterpart. This work gives the first convergence analysis of the DP deep learning, through the lens of training dynamics and the neural tangent kernel (NTK) matrix. Our convergence theory successfully characterizes the effects of two key components in the DP training: the per-sample clipping and the noise addition. We initiate a general principled framework to understand the DP deep learning with any network architecture, loss function and various optimizers including DP-Adam. Our analysis also motivates a new clipping method, the *global clipping*, that significantly improves the convergence, as well as preserves the same DP guarantee and computational efficiency as the existing method, which we term as *local clipping*. In addition, our global clipping is surprisingly effective at learning *calibrated classifiers*, in contrast to the existing DP classifiers which are oftentimes over-confident and unreliable. Implementation-wise, the new clipping can be realized by inserting one line of code into the Pytorch `Opacus` library.

## 1 Introduction

Deep learning has achieved tremendous success in many applications that involve crowd-sourced information, e.g., face image, emails, financial status, and medical records. However, using such sensitive data raises severe privacy concerns on a range of image recognition, natural language processing and other tasks (Cadwalladr & Graham-Harrison, 2018; Rocher et al., 2019; Ohm, 2009; De Montjoye et al., 2013; 2015). For concrete examples, there are multiple successful privacy attacks on deep learning models, in which members in the dataset can be re-identified using the location or the purchase record, via the membership inference attack (Shokri et al., 2017; Carlini et al., 2019). In another example, the attackers can extract a person's name, email address, phone number, and physical address from the billion-parameter GPT-2 (Radford et al., 2019) via the extraction attack (Carlini et al., 2020). Therefore, many studies have applied differential privacy (DP) (Dwork et al., 2006; Dwork, 2008; Dwork et al., 2014; Mironov, 2017; Duchi et al., 2013; Dong et al., 2019), a mathematically rigorous approach, to protect against leakage of private information (Abadi et al., 2016; McSherry & Talwar, 2007; McMahan et al., 2017; Geyer et al., 2017). To achieve this gold standard of privacy guarantee, since the seminal work (Abadi et al., 2016), DP optimizers are applied to train the neural networks while preserving the accuracy of prediction. To name a few, researchers have proposed DP-SGD (Abadi et al., 2016; Bassily et al., 2014), DP-Adam (Bu et al., 2019), DP-SGLD (Wang et al., 2015; Li et al., 2019), DP-FTRL (Kairouz et al., 2021), DP-FedSGD and DP-FedAvg (McMahan et al., 2017).

Algorithmically speaking, DP optimizers generally have two extra steps in comparison to non-DP standard optimizers: the per-sample clipping and the random noise addition, so that DP optimizers descend in the direction of the averaged, clipped, noisy gradient (see Figure 1). These extra steps protect the resulting models against privacy attacks via the Gaussian mechanism (Dwork et al., 2014, Theorem A.1), at the expense of an empirical performance degradation compared to the non-DP deep learning, in terms of much slower convergence and lower utility. For example, state-of-the-art CIFAR10 accuracy with DP is $\approx 70\%$ without pre-training (Papernot et al., 2020) (while the same non-DP networks can

achieve 90% accuracy) and similar performance drops have been observed on facial images, tweets, and many other datasets (Bagdasaryan et al., 2019).

Empirically, many works have evaluated the effects of noise scale, batch size, clipping norm, learning rate, and network architecture on the privacy-accuracy trade-off (Abadi et al., 2016; Papernot et al., 2020). However, despite the prevalent usage of DP optimizers, little is known about its convergence behavior from a theoretical viewpoint, which is necessary to understand and improve the deep learning with differential privacy. We notice one previous attempt by (Chen et al., 2020), analyzing the DP-SGD convergence with an assumption of symmetric gradient distribution, which can be unrealistic and inapplicable to real datasets.

**Our Contributions**

- We explicitly characterize the *general training dynamics* of deep learning with DP gradient methods (e.g., DP-GD and DP-Adam). We show a fundamental influence of the DP training on the NTK matrix, which causes the convergence to worsen. This analysis leads to a *convergence theory* for the DP deep learning.
- We propose a novel principle for designing the DP optimizers and thus develop a new *global clipping* method that provably enjoys desirable convergence behaviors.
- We demonstrate via numerous experiments that DP optimizers with global clipping significantly improve the loss convergence. Interestingly, our clipping further effectively mitigates the *calibration* issue of existing DP classifiers, which usually exacebates the "over-confidence" in non-DP models.
- Our global clipping has the *same privacy guarantee* and *computational efficiency* as the local clipping, which leads to a *mix-up training* strategy where local and global clippings are applied interchangeably.
- Our global clipping is *easy-to-code* (see Appendix D) and *generalizable* to arbitrary optimizers, network architectures, loss functions, and tasks including federated learning.

A quick preview of the comparison among the DP optimizers with the local and the global clipping is as follows:

| Clipping type | Positive NTK | Loss convergence | Monotone loss decay | To zero loss |
|---|---|---|---|---|
| No clipping | Yes | Yes | Yes | Yes |
| Local & Flat | No | No | No | Yes |
| Local & Layerwise | No | No | No | No |
| Global & Flat | Yes | Yes | Yes | Yes |
| Global & Layerwise | Yes | Yes | Yes | Yes |

Table 1: Effects of different per-sample clippings on deep learning with DP-GD, assuming no screening happens in global clipping. Here "Yes/No" means guaranteed or not and the loss refers to the training set. "Loss convergence" is conditioned on $\mathbf{H}(t) \succ 0$ (see (2.1)).

## 2 Warmup: Convergence of Non-Private Gradient Descent

We start by reviewing the standard, non-DP Gradient Descent (GD) for **arbitrary neural network** and **arbitrary loss**, before we dive into the analysis of DP optimizers. In particular, we analyze the training dynamics using the neural tangent kernel (NTK) matrix[1].

Suppose a neural network $f$ is governed by weights $\mathbf{w}$, with samples $\boldsymbol{x}_i$ and labels $y_i$ ($i = 1, ..., n$). Denote the prediction by $f_i = f(\boldsymbol{x}_i, \mathbf{w})$, and the per-sample loss by $\ell_i = \ell(f(\boldsymbol{x}_i, \mathbf{w}), y_i)$ for some loss function $\ell$. We define the objective function $L$ to be the average of per-sample losses $L(\mathbf{w}) = \frac{1}{n} \sum_{i=1}^{n} \ell(f(\boldsymbol{x}_i, \mathbf{w}), y_i)$. The discrete gradient descent with a step size $\eta$, and the corresponding *gradient flow*[2] are:

$$\mathbf{w}(k+1) = \mathbf{w}(k) - \eta \frac{\partial L}{\partial \mathbf{w}}^{\top}, \text{ and } \dot{\mathbf{w}}(t) = -\frac{\partial L}{\partial \mathbf{w}}^{\top} = -\frac{1}{n}\sum_i \nabla_{\mathbf{w}}\ell_i(t).$$

---

[1]We emphasize that our analysis works on any neural networks, not limited to the infinitely wide or over-parameterized ones. Put differently, we don't assume the NTK matrix $\mathbf{H}$ to be deterministic nor nearly time-independent, as was the case in (Arora et al., 2019a; Lee et al., 2019; Du et al., 2018; Allen-Zhu et al., 2019; Zou et al., 2020; Fort et al., 2020; Arora et al., 2019b).

[2]I.e., the ordinary differential equation (ODE) describing the weight updates with infinitely small step size $\eta \to 0$ in the continuous time.

Applying the chain rules, we obtain the following general dynamics of the loss $L$,

$$\dot{L} = \frac{\partial L}{\partial \mathbf{w}} \dot{\mathbf{w}} = -\frac{\partial L}{\partial \mathbf{w}} \frac{\partial L}{\partial \mathbf{w}}^\top = -\frac{\partial L}{\partial \boldsymbol{f}} \frac{\partial \boldsymbol{f}}{\partial \mathbf{w}} \frac{\partial \boldsymbol{f}}{\partial \mathbf{w}}^\top \frac{\partial L}{\partial \boldsymbol{f}}^\top = -\frac{\partial L}{\partial \boldsymbol{f}} \mathbf{H}(t) \frac{\partial L}{\partial \boldsymbol{f}}^\top , \qquad (2.1)$$

where $\frac{\partial L}{\partial \boldsymbol{f}} = \frac{1}{n}(\frac{\partial \ell_1}{\partial f_1}, ..., \frac{\partial \ell_n}{\partial f_n}) \in \mathbb{R}^{1 \times n}$, and the Gram matrix $\mathbf{H}(t) := \frac{\partial \boldsymbol{f}}{\partial \mathbf{w}} \frac{\partial \boldsymbol{f}}{\partial \mathbf{w}}^\top \in \mathbb{R}^{n \times n}$ is known as the NTK matrix, which is positive semi-definite and crucial to analyzing the convergence behavior. To give a concrete example, let $\ell$ be the MSE loss $\ell_i(\mathbf{w}) = (f(\boldsymbol{x}_i, \mathbf{w}) - y_i)^2$ and $L_{\text{MSE}} = \frac{1}{n} \sum_i (f_i - y_i)^2$, then $\dot{L}_{\text{MSE}} = -4(\boldsymbol{f} - \boldsymbol{y})^\top \mathbf{H}(t)(\boldsymbol{f} - \boldsymbol{y})/n^2$. Furthermore, if $\mathbf{H}(t)$ is positive definite, the MSE loss $L_{\text{MSE}} \to 0$ exponentially fast (Du et al., 2018; Allen-Zhu et al., 2019; Zou et al., 2020) , the cross-entropy loss $L_{\text{CE}} \to 0$ at rate $O(1/t)$ and any loss convex in the prediction $L = \sum_i \ell_i/n$ converges to 0 (Allen-Zhu et al., 2019).

# 3 DIFFERENTIALLY PRIVATE GRADIENT METHODS AND GLOBAL CLIPPING

We now introduce the DP optimizers (Google; Facebook) to train the DP neural networks. One popular optimizer is the DP-SGD (Song et al., 2013; Chaudhuri et al., 2011; Abadi et al., 2016; Bu et al., 2019) in Algorithm 1 and more optimizers such as DP-Adam and DP-FedAvg (McMahan et al., 2017) for federated learning can be found in Appendix F. In contrast to the standard SGD, the DP-SGD has two unique steps: the *per-sample clipping* (to bound the sensitivity of per-sample gradients) and the random *noise addition* (to guarantee the privacy of models), both are discussed in details via the Gaussian mechanism in Lemma B.1.

---

**Algorithm 1** DP-SGD (with local or global flat per-sample clipping)

---

**Parameters:** initial weights $\mathbf{w}_0$, learning rate $\eta_t$, subsampling probability $p$, number of iterations $T$, noise scale $\sigma$, gradient norm bound $R$, maximum norm bound $Z \geq R$.

    **for** $t = 0, \ldots, T - 1$ **do**

        Subsample a batch $I_t \subseteq \{1, \ldots, n\}$ from training set with probability $p$

        **for** $i \in I_t$ **do**

            $v_t^{(i)} \leftarrow \nabla_{\mathbf{w}} \ell(f(\boldsymbol{x}_i, \mathbf{w}_t), y_i)$

            Option 1: $C_{local,i} = \min\left\{1, R/\|v_t^{(i)}\|_2\right\}$        ▷ Local clipping factor (existing)

            Option 2: $C_{global,i} \equiv \begin{cases} R/Z & \text{if } \|v_t^{(i)}\|_2 \leq Z \\ 0 & \text{if } \|v_t^{(i)}\|_2 > Z \end{cases}$   ▷ Global clipping factor (ours)

            $\bar{v}_t^{(i)} \leftarrow C_i \cdot v_t^{(i)}$                       ▷ Clip the gradient

        $\bar{V}_t \leftarrow \sum_{i \in I_t} \bar{v}_t^{(i)}$                      ▷ Sum over batch

        $\mathbf{w}_{t+1} \leftarrow \mathbf{w}_t - \frac{\eta_t}{|I_t|}\left(\bar{V}_t + \sigma R \cdot \mathcal{N}(0, I)\right)$   ▷ Apply Gaussian mechanism and descend

---

Although the per-sample clipping is widely applied in DP deep learning, its effect on convergence remains a mystery. Empirical observations have found that optimizers with the per-sample clipping (even when no noise is present) have much worse convergence and accuracy (Abadi et al., 2016; Bagdasaryan et al., 2019). In fact, the current form of clipping is heuristic and lacks theoretical understanding, especially when the noise addition is present. In what follows, we use $C$ to denote $C_{local}$ or $C_{global}$ when it is clear from the context.

We propose and analyze a new clipping, namely the **global clipping** (see Option 2 in Algorithm 1), where the clipping operation takes place on all per-sample gradients that pass the screening procedure. From this viewpoint, the global clipping is a *batch clipping* instead of an individual clipping (see Appendix F.6 for comparison with local clipping). More precisely, in local clipping, each per-sample gradient $\nabla_{\mathbf{w}} \ell_i$ compare its length to $R$ and multiplied with a sample-specific clipping factor $0 < C_i \leq 1$. In global clipping, only $\nabla_{\mathbf{w}} \ell_i$ with norm smaller than $Z$ is used (otherwise $C_i = 0$) and multiplied with a common clipping factor $R/Z$, which guarantees the sensitivity to be $R$ as in local clipping. At a high level, the idea of global clipping is to preserve the gradient direction (i.e. to remove the gradient bias) while bounding the sensitivity during the clipping, which will guarantee the positive semi-definiteness of the NTK matrix via Theorem 2.

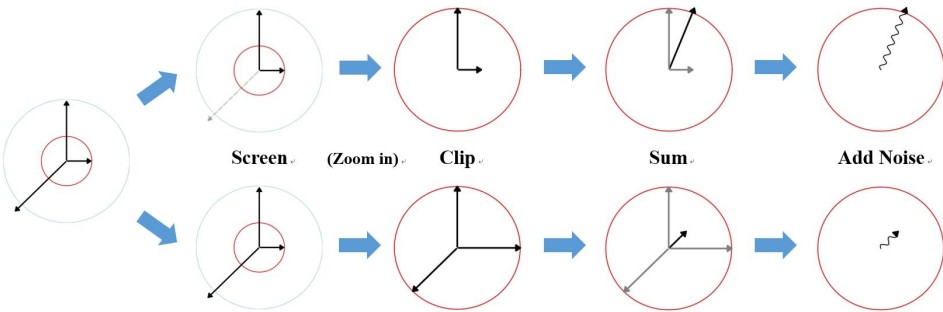

Figure 1: Illustration of global (upper) and local per-sample clipping (lower) in Algorithm 1. Black arrowed lines are per-sample gradients. The circles have radius $R$ (red) and $Z$ (grey).

## 4 Convergence Analysis of DP Optimizers

In this section, we analyze the weight and loss dynamics of DP optimizers with the local or global per-sample clipping, denoted in the subscript, e.g., DP-SGD$_{local}$ and DP-SGD$_{global}$. Our narrative here focuses on the widely used DP-GD for the sake of simplicity, yet and our analysis generalizes to other full-batch DP optimizers such as DP-HeavyBall, DP-RMSprop, and DP-Adam as well (see Theorem 4 and Appendix F).

### 4.1 Effect of Noise Addition on Convergence

Our first result is easy yet surprising: the gradient flow of a stochastic noisy GD with non-zero noise (4.1) is the same as that of a deterministic dynamics without the noise (4.2). Put differently, the noise addition has no effect on the convergence of DP optimizers in the continuous time gradient flow. This is a common phenomenon called *certainty equivalence* in the stochastic control community with the name of (Chow et al., 1975).

To elaborate this point, we consider the DP-GD with Gaussian noise, as in Algorithm 1,

$$\mathbf{w}(k+1) = \mathbf{w}(k) - \frac{\eta}{n}\Big(\sum_i \nabla_{\mathbf{w}}\ell_i C_i + \sigma R \cdot \mathcal{N}(0,1)\Big). \tag{4.1}$$

Notice that this general dynamics covers both the non-DP GD ($\sigma = 0$ and $C_i \equiv 1$) and DP-GD with local or global clipping. Through Fact 4.1, we claim that the gradient flow of (4.1) is the same ODE regardless of the value of $\sigma$, whose proof is delayed to Appendix B.

**Fact 4.1.** For all $\sigma \geq 0$, the gradient descent in (4.1) has the continuous gradient flow

$$d\mathbf{w}(t) = -\frac{1}{n}\sum_i \nabla_{\mathbf{w}}\ell_i(t)C_i(t)dt. \tag{4.2}$$

This result shares the spirit of the conventional wisdom[3] that tune the clipping norm $C$ first (e.g. setting $\sigma = 0$ or small), and tune the noise level $\sigma$ afterwards, since the convergence is more sensitive to the clipping factor. We visualize this point via experiment in Appendix G.

**Remark 4.2.** Our proof of Fact 4.1 generally holds true for any DP optimizer besides DP-GD: as $\eta \to 0$, different $\sigma$ result in the same gradient flow.

### 4.2 Effect of Per-Sample Clipping on NTK Matrix

We move on to analyze the effect of the per-sample clipping on the DP training (4.2). It has been empirically observed that the per-sample clipping results in a worse convergence and accuracy even without the noise (Bagdasaryan et al., 2019). We highlight that the NTK matrix is the key to understand the convergence behavior, and that the clipping affects NTK through its linear algebra properties, especially the positive semi-definiteness, which we define below in two notions for a *general* matrix.

**Definition 4.3.** For a (not necessarily symmetric) matrix $A$, it is

1. *positive in quadratic form* if and only if $\mathbf{x}^\top A\mathbf{x} \geq 0$ for every non-zero $\mathbf{x}$;

2. *positive in eigenvalues* if and only if all eigenvalues of $A$ are non-negative.

---

[3]See https://github.com/pytorch/opacus/blob/master/tutorials/building_image_classifier.ipynb

These two positivity definitions are equivalent for a symmetric or Hermitian matrix, but not so for non-symmetric matrices. We illustrate this difference in Appendix A with some concrete examples. Next, we introduce two styles of per-sample clippings. Both can be implemented locally or globally.

**Flat Clipping** The DP-GD described in Algorithm 1 and (4.1), with the gradient flow (4.2), is equipped with the *flat* clipping (McMahan et al., 2018). In words, the flat clipping upper bounds the entire gradient vector by a single $R$. Using the chain rules, we get

$$\dot{L} = \frac{\partial L}{\partial \mathbf{w}} \dot{\mathbf{w}} = -\frac{1}{n^2} \sum_j \nabla_{\mathbf{w}} \ell_j \sum_i \nabla_{\mathbf{w}} \ell_i C_i = -\frac{\partial L}{\partial \boldsymbol{f}} \mathbf{H} \mathbf{C} \frac{\partial L}{\partial \boldsymbol{f}}^\top , \qquad (4.3)$$

where $\mathbf{C}(t) = \mathrm{diag}(C_1, \cdots, C_n)$ is the clipping matrix, with $C_i$ defined in Algorithm 1.

**Layerwise Clipping.** We additionally analyze another widely used clipping – the *layerwise* clipping (Abadi et al., 2016; McMahan et al., 2017; Phan et al., 2017). Unlike the flat clipping, the layerwise clipping upper bounds the $r$-th layer's gradient vector by a layer-dependent norm $R_r$, as demonstrated in Algorithm 2. Hence $\dot{L} = -\sum_r \frac{\partial L}{\partial \boldsymbol{f}} \mathbf{H}_r \mathbf{C}_r \frac{\partial L}{\partial \boldsymbol{f}}^\top$, where the layerwise NTK matrix $\mathbf{H}_r = \frac{\partial \boldsymbol{f}}{\partial \mathbf{w}_r} \frac{\partial \boldsymbol{f}}{\partial \mathbf{w}_r}^\top$, and $\mathbf{C}_r(t) = \mathrm{diag}(C_{1,r}, \cdots, C_{n,r})$.

### 4.3 LOCAL PER-SAMPLE CLIPPING BREAKS NTK POSITIVITY

We start with the analysis of local clipping, which is the prevailing clipping technique prior to our work. We show that the DP-GD with local clipping breaks the traditional positive semi-definiteness of the NTK matrix[4].

**Theorem 1.** *For an arbitrary neural network and a loss convex in $f$, suppose we clip the per-sample gradients **locally**, and assume $\mathbf{H}(t) \succ 0$, then in the gradient flow of DP-GD:*

1. *The local flat clipping has the loss dynamics in (4.3), with NTK matrix $\mathbf{H}(t)\mathbf{C}_{local}(t)$, which may not be symmetric nor positive in quadratic form, but is positive in eigenvalues.*

2. *The local layerwise clipping has the loss dynamics with NTK matrix $\sum_r \mathbf{H}_r(t)\mathbf{C}_{local,r}(t)$, which may not be symmetric nor positive in quadratic form or in eigenvalues.*

3. *For both local flat and layerwise clipping, the loss $L(t)$ may not decrease monotonically.*

4. *If the loss $L(t)$ converges, for the flat clipping, it converges to 0; for the layerwise clipping, it may converge to a non-zero value.*

We prove Theorem 1 in Appendix B. The theorem states that the symmetry of NTK is almost surely broken by the local clipping. In that case, severe issues arise in the loss convergence, which are depicted in Figure 5 and Figure 7.

### 4.4 GLOBAL PER-SAMPLE CLIPPING PRESERVES NTK POSITIVITY WITH LARGE $Z$

Now we switch gears to our global clipping. At each iteration when $Z$ is sufficiently large so that no per-sample gradient is screened out, the global clipping clearly corresponds to a symmetric and positive semi-definite NTK matrix $\mathbf{H}(t)C(t)$ in flat clipping and $\sum_r \mathbf{H}_r(t)C_r(t)$ in layerwise clipping, since all per-sample gradients share the same clipping factor. As a result, the clipping matrices are indeed scalar in that $\mathbf{C} = C\mathbf{I}$ in (4.3) and $\mathbf{C}_r = C_r\mathbf{I}$ in (B.1). Hence we obtain the following result for the global clipping.

**Theorem 2.** *For an arbitrary neural network and a loss convex in $f$, suppose we clip the per-sample gradients **globally**, assuming $\mathbf{H}(t) \succ 0$ and $\|v_t^{(i)}\|_2 \leq Z$,[5] then in the gradient flow of DP-GD:*

1. *The global flat (resp. layerwise) clipping has loss dynamics in (4.3), with NTK matrix $\mathbf{H}(t)C_{global}(t)$ (resp. $\sum_r \mathbf{H}_r(t)C_{global,r}(t)$), which is symmetric and positive definite.*

2. *For both global flat and layerwise clipping, the loss $L(t)$ decreases monotonically to 0.*

---

[4]It is a fact that the product of a symmetric positive definite matrices and a positive diagonal matrix may not be symmetric nor positive in quadratic form. This is shown in Appendix A.

[5]If $Z$ is not large and the screening is effective, then the global clipping (flat or layerwise) may break its symmetry and positivity both in quadratic form and in eigenvalues. Consequently, the training loss may not decrease monotonically nor to zero.

We prove Theorem 2 in Appendix B and the benefits of the global clipping are assessed in Section 6. Our findings from Theorem 1 and Theorem 2 are visualized in the left plot of Figure 9 and summarized in Table 2, which further leads to Table 1.

| Clipping method | NTK matrix | Symmetric matrix | Positive in quadratic form | Positive in eigenvalues |
|---|---|---|---|---|
| No clipping | $\mathbf{H} \equiv \sum_r \mathbf{H}_r$ | Yes | Yes | Yes |
| Local & Flat | $\mathbf{H}\mathbf{C}$ | No | No | Yes |
| Local & Layerwise | $\sum_r \mathbf{H}_r \mathbf{C}_r$ | No | No | No |
| Global & Flat | $\mathbf{H}C$ | Yes | Yes | Yes |
| Global & Layerwise | $\sum_r \mathbf{H}_r C_r$ | Yes | Yes | Yes |

Table 2: Linear algebra properties of NTK by different clipping methods, assuming no screening happens in global clipping. Here 'Yes/No' means guaranteed or not.

## 5 Privacy Analysis of DP Optimizers

In this section we define DP and prove that DP optimizers using the global clipping have the **same** privacy guarantee as those using the local clipping. Notice that for the privacy analysis, we work with the general DP optimizers, including those with mini-batches.

**Definition 5.1.** A randomized algorithm $M$ is $(\varepsilon, \delta)$-differentially private (DP) if for any neighboring datasets $S, S'$ differ by an arbitrary sample, and for any event $E$,

$$\mathbb{P}[M(S) \in E] \leqslant e^{\varepsilon}\mathbb{P}\left[M\left(S'\right) \in E\right] + \delta.$$

A common approach to guarantee DP when approximating a function $g$ is via additive noise calibrated to $g$'s sensitivity (Dwork et al., 2006). This is known as the Gaussian mechanism and widely used in DP deep learning, see more details in Lemma B.1.

For the same differentially private mechanism, different privacy accountants (e.g., Moments accountant (Abadi et al., 2016; Canonne et al., 2020), Gaussian differential privacy (GDP) (Dong et al., 2019; Bu et al., 2019), Fourier accountant (Koskela et al., 2020), each based on a different composition theory) accumulate the privacy risk $\epsilon(\sigma, n, p, \delta, T)$ differently over $T$ iterations. The next result shows that DP optimizers with global clipping is as private as those with local clipping, independent of the choice of the privacy accountant.

**Theorem 3.** *DP optimizers with the local or global clipping are equally $(\epsilon, \delta)$-DP.*

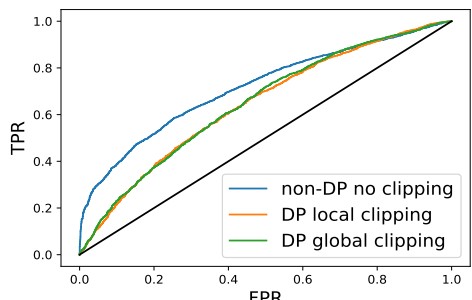

Figure 2: Attack model's ROC-AUC on entire CIFAR10 in Section 6.2. non-DP AUC, 0.717; DP-SGD$_{local}$, 0.644; DP-SGD$_{global}$, 0.648.

While a DP model by definition is resilient to all types of privacy attacks, we illustrate that DP-SGD$_{global}$ offers similar privacy protection to DP-SGD$_{local}$ against the membership inference attacks (MIA) in Figure 2. MIA is a common privacy attack by which the attacker aims to determine whether a given data point belongs to the sensitive training set [26, 38, 41, 48]. In our setting, the black-box attacker uses a logistic regression that only has access to the prediction logits and labels. The privacy vulnerability is characterized as the attack model's AUC, while lower AUC is preferred.

## 6 Numerical Results

We highlight that the global clipping works with any DP optimizers (e.g., DP-Adam, DP-RMSprop, DP-FTRL(Kairouz et al., 2021), DP-SGD-JL(Bu et al., 2021a), etc.) that employ the local clipping, with *almost identical computational complexity* (discussed in Appendix D). Empirically, DP optimizers with global clipping improve over existing DP optimizers on the convergence of training and generalization losses. We thus reveal a novel phenomenon that DP optimizers play important roles in producing well-calibrated and reliable models. For all experiments, we use the GDP privacy accountant, with Pytorch `Opacus` library and on a Google Colab P100 GPU. More details are available in Appendix E.

In $M$-class classification problems, we denote the probability prediction for the $i$-th sample as $\boldsymbol{\pi}_i \in \mathbb{R}^M$ so that $f(\boldsymbol{x}_i) = \text{argmax}(\boldsymbol{\pi}_i)$, then the accuracy is $\mathbf{1}\{f(\boldsymbol{x}_i) = y_i\}$. The confidence, i.e., the probability associated with the predicted class or maximum softmax probability, is $\hat{P}_i := \max_{k=1}^M [\boldsymbol{\pi}_i]_k$ and a good calibration means the confidence is close to the accuracy[6]. Formally, we employ two popular calibration metrics from (Naeini et al., 2015) in Table 3: the Expected Calibration Error (ECE) and the Maximum Calibration Error (MCE)

$$\text{ECE: } \mathbb{E}_{\hat{P}_i}\left[\left|\mathbb{P}(f(\boldsymbol{x}_i) = y_i | \hat{P}_i = p) - p\right|\right], \quad \text{MCE: } \max_{p \in [0,1]} \left|\mathbb{P}(f(\boldsymbol{x}_i) = y_i | \hat{P}_i = p) - p\right|.$$

|  | ECE % | | | MCE % | | |
|---|---|---|---|---|---|---|
|  | non-DP | DP local | DP global | non-DP | DP local | DP global |
| CIFAR10 | 13.9 | 20.0 | 3.3 | 20.9 | 32.0 | 9.9 |
| SNLI | 13.0 | 22.0 | 17.6 | 34.7 | 62.5 | 28.9 |
| MNIST | 0.8 | 2.5 | 0.5 | 21.1 | 50.2 | 22.8 |

Table 3: Calibration metrics ECE and MCE by non-DP (no clipping) and DP optimizers. Note that the SNLI's DP global stands for mix-up training described in Section 6.3.

### 6.1 MNIST IMAGE DATA WITH CNN MODEL

On the MNIST dataset, which contains 60000 training samples and 10000 test samples of $28 \times 28$ grayscale images in 10 classes, we use the standard CNN in the DP libraries[7](Google; Facebook) (see Appendix E.1 for architecture) and train with DP-SGD. In Figure 3, both clippings result in $(2.32, 10^{-5})$-DP, similar test accuracy (96% for local and 95% for global), though the global clipping leads to smaller loss. In right sub-plot of Figure 3, we demonstrate how $Z$ affects the performance of global clipping, ceteris paribus.

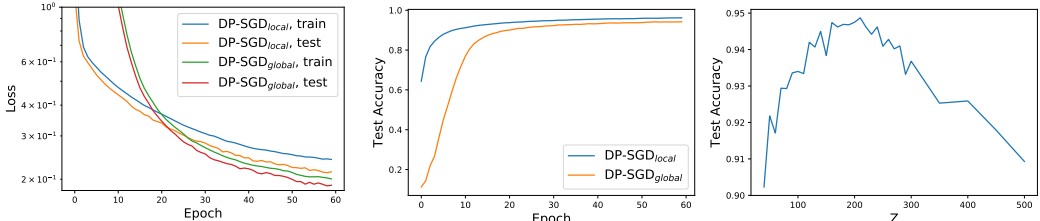

Figure 3: Loss (left) and accuracy (right) on MNIST with 4-layer CNN under different clipping methods, batch size 256, learning rate 0.15, noise scale 1.1, clipping norm 1.0; for global clipping, we choose $Z = 210$ as the maximum gradient bound, $(\epsilon, \delta) = (2.32, 10^{-5})$.

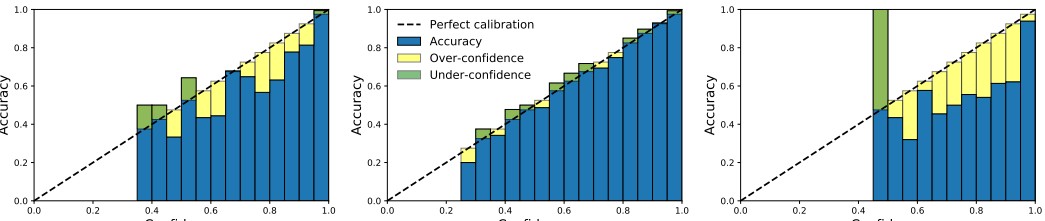

Figure 4: Reliability diagrams (left for non-DP; middle for global clipping; right for local clipping) on MNIST with 4-layer CNN.

In Figure 4, the *reliability diagram* (DeGroot & Fienberg, 1983; Niculescu-Mizil & Caruana, 2005) displays the accuracy as a function of confidence. Graphically speaking, a calibrated classifier is expected to have blue bins close to the diagonal black dotted line. While the non-DP model is generally over-confident and thus not calibrated, the global clipping effectively achieves nearly perfect calibration. In contrast, the classifier with local clipping is not only mis-calibrated, but is 'bipolar disordered': it is either over-confident and inaccurate, or under-confident but highly accurate. This is observed in all classification experiments.

---

[6]Over-confident classifiers, with wrong prediction at one data point, reduce accuracy a little but increase loss significantly due to large $\log(\pi_{y_i})$, since small probability is assigned to true class.

[7]See https://github.com/tensorflow/privacy/tree/master/tutorials in Tensorflow and https://github.com/pytorch/opacus/blob/master/examples/mnist.py in Pytorch Opacus.

## 6.2 CIFAR10 IMAGE DATA WITH CNN MODEL

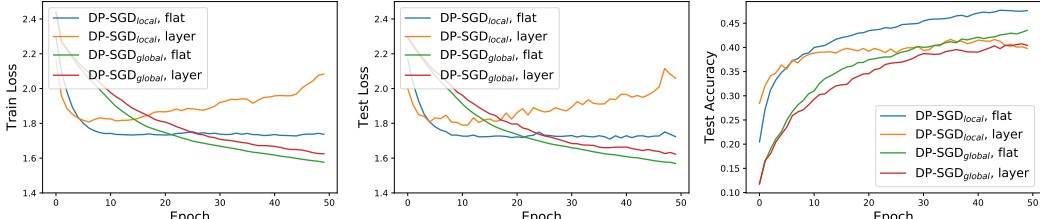

Figure 5: Loss (left and middle) and accuracy (right) on CIFAR10 with 5-layer CNN under different clipping methods, batch size 250, learing rate 0.05, noise scale 1.3, $Z = 75$, clipping norm 1.5 (flat). For layerwise clipping, global: $[1.5, 0.3]$ per layer (1.5 for weights, 0.3 for biases); local: $[1.5, 1.5]$, $(\epsilon, \delta) = (1.96, 10^{-5})$.

CIFAR10 is a more challenging image dataset, which contains 50000 training samples and 10000 test samples of $32 \times 32$ color images in 10 classes. We use the standard CNN on Pytorch CIFAR10 tutorial[8] (see Appendix E.2 for architecture) and train with DP-SGD without pre-training (unlike (Abadi et al., 2016; Xu et al., 2020), which pretrain on CIFAR100). Both clippings result in $(1.96, 10^{-5})$-DP and the test accuracy (local: 47.6%; global: 43.5%; non-DP: 61.3%) is comparable with state-of-the-art in (Papernot et al., 2020), which is around 47% at this privacy budget. Clearly from Figure 5, global clipping has better convergence and similar accuracy than local clipping. Especially, local layerwise clipping can be unstable, as indicated by Theorem 1. From Figure 6, we can clearly see that DP-SGD$_{local}$ results in poorly calibrated classifiers on CIFAR10 but DP-SGD$_{global}$ is well-calibrated.

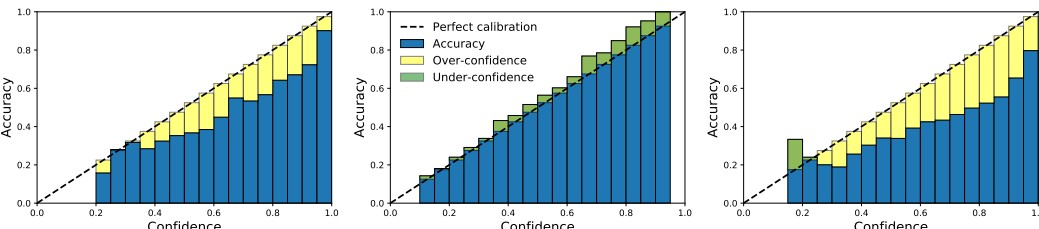

Figure 6: Reliability diagrams (left for non-DP; middle for global clipping; right for local clipping) on CIFAR10 with 5-layer CNN.

## 6.3 SNLI TEXT DATA WITH BERT AND MIX-UP TRAINING

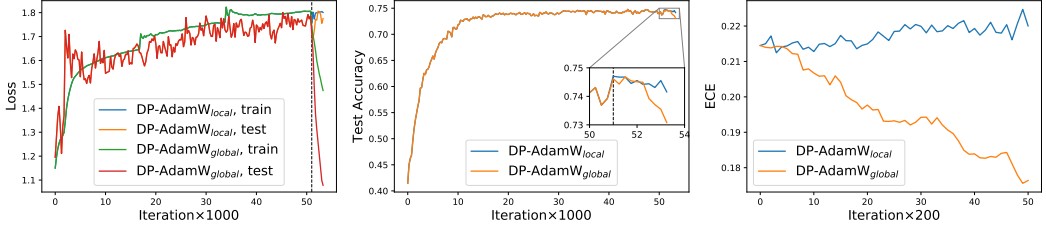

Figure 7: Loss (left), accuracy (middle) and calibration after switching clipping (right) on SNLI with pre-trained BERT, batch size 32, learning rate 0.0005, noise scale 0.4, $Z = 1000$, clipping norm 0.1, $(\epsilon, \delta) = (1.25, 1/550152)$.

Stanford Natural Language Inference (SNLI) [9] is a collection of human-written English sentence paired with one of three classes: entailment, contradiction, or neutral. The dataset has 550152 training samples and 10000 test samples. We use the pre-trained BERT (Bidirectional Encoder Representations from Transformers) on `Opacus` tutorial[10], which gives a state-of-the-art privacy-accuracy result. Our BERT contains 108M parameters and we only train the last Transformer encoder, which has 7M parameters, using DP-AdamW. In

---

[8]See `https://pytorch.org/tutorials/beginner/blitz/cifar10_tutorial.html`.

[9]We use SNLI 1.0 from `https://nlp.stanford.edu/projects/snli/`

[10]See github `pytorch/opacus/blob/master/tutorials/building_text_classifier.ipynb`.

particular, we use a **mix-up training**: for global clipping, we in fact train BERT with DP-$\text{SGD}_{local}$ for 3 epochs ($51.5 \times 10^3$ iterations) and then use DP-$\text{SGD}_{global}$ for an additional 2500 iterations. In other words, 95% of the training is done with local clipping but the last 5% is done with global clipping. For local clipping, DP-$\text{SGD}_{local}$ is used for the entire training process of 54076 iterations.

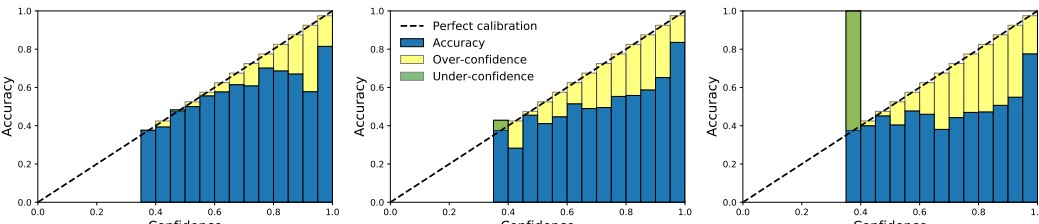

Figure 8: Reliability diagrams (left for non-DP; middle for global clipping; right for local clipping) on SNLI with BERT. Note that global clipping is only used for the last 2500 iterations out of the entire 54000 iterations.

Surprisingly, the existing DP optimizer does not minimize the loss at all, yet the accuracy still improves along the training. We again observe that global clipping has significantly better convergence than the local clipping (observe that when turned to global clipping in the last 2500 steps, the test loss decreases significantly from 1.79 to 1.08, and the training loss decreases from 1.81 to 1.47; while keeping local clipping has no effect on reducing the losses). The resulting global model also has similar accuracy (local: 74.1%; global: 73.1%; as a benchmark, non-DP: 85.4%), same privacy ($\epsilon = 1.25, \delta = 1/550152$), and much better calibration in comparison to the local clipping (see Table 3). We remark that all hyperparameters are the same as in the `Opacus` tutorial.

### 6.4 REGRESSION TASKS

On regression tasks, the performance measure and the loss function are unified as MSE. Figure 9 shows that global clipping is comparable if not better than local clipping. We experiment on the California Housing data (20640 samples, 8 features) and Wine Quality (1599 samples, 11 features, run with full-batch DP-GD). Additional experimental details are available in Appendix E.4.

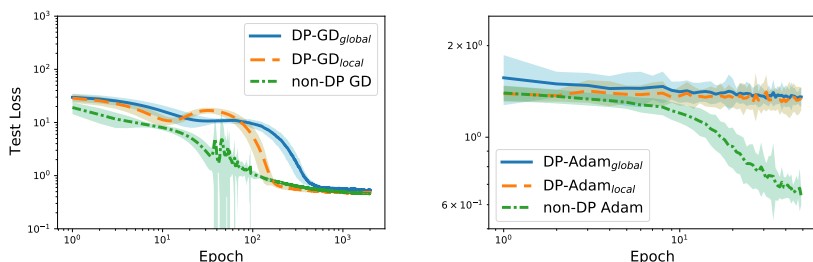

Figure 9: Performance of DP optimizers under different clipping methods on the Wine Quality with $Z = 400$ (left) and the California Housing datasets with $Z = 2000$ (right).

## 7 DISCUSSION

In this paper, we establish a framework of the convergence analysis for DP deep learning, via the NTK matrix, that applies to general neural network architecture, loss function, and optimization algorithm. We show that in the continuous time analysis, the noise addition does not affect the convergence but the per-sample clipping does. We then propose the global clipping method, which has provable advantages in convergence with the same privacy guarantee and efficiency as the existing local clipping. Hence, one may apply two clippings interchangeably during the mix-up training. Our global clipping significantly outperform the local clipping in terms of loss and better calibration. Future directions include the discrete time analysis as well as mini-batches. This means that the added noise and the sub-sampling noise will come into effect, and requires analysis of stochastic differential equation.

## 8 ETHICS STATEMENT

The experiments in this work is conducted on publically available datasets. The methods in this paper should not raise ethical concerns.

## 9 REPRODUCIBILITY STATEMENT

Our code is easily reproducible, since we have already provide the full implementation in the Appendix D. For the script that reproduces our experiments, we have submitted a set of template code that covers a large portion of our experiments. For the details of our our datasets, we describe in details in section 6 and Appendix E. We state our assumptions clearly in all our theoretical results, especially in the theorem statements.

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
