# OpenReview forum: "On the Convergence and Calibration of Deep Learning with Differential Privacy"
_ICLR.cc/2022/Conference — ICLR 2022 Submitted_

### Official Review · Reviewer_Yf7v · 2021-11-01

**Correctness:** 4
**Technical Novelty And Significance:** 4
**Empirical Novelty And Significance:** 4
**Recommendation:** 8
**Confidence:** 4

**Main Review:**

- Simple and elegant clipping
- Nice theoretical analysis
- Nice empirical showing

**Summary Of The Paper:**

The paper proposes a new clipping method for DP-SGD. It is shown that this new clipping method performs favorably empirically. Furthermore, the paper analyzes the clipping method using NTK.

**Summary Of The Review:**

I am in favor of this work. The global clipping is simple but elegant. Furthermore, the NTK analysis is nice and helps support the intuition behind the method. I think that using NTK to analyze differential privacy is actually a method with potentially widespread applicability. Furthermore, the empirical analysis of the method seems sound. The issue of calibration with DP-SGD is also interesting and understudied. I find few faults in this work, and the brevity of this review is a consequence of having little to critique.

---

> ### Author Response · Authors · 2021-11-13
> **Response to reviewer Yf7v**
>
> We would like to extend our gratitude towards the reviewer for the appreciation of our work! We are also highly in favor of this work and believe it conveys some nice intuition and concise insight as of the DP optimizers and the effect of clipping. We would like to answer any further questions from the reviewer had any come up. We are also working on a follow-up to better understand the calibration phenomenon better from a theoretical viewpoint.

---

### Official Review · Reviewer_b1Rw · 2021-11-01

**Correctness:** 2
**Technical Novelty And Significance:** 2
**Empirical Novelty And Significance:** 2
**Recommendation:** 3
**Confidence:** 4

**Main Review:**

### ***Overall merit of the paper.***

The problem of better understanding the properties of differentially private optimizers is an important topic within the ML community, as it would allow building more robust techniques with provable convergence guarantees. Moreover, the possibility of analyzing the dynamics using the continuous gradient flow seems to be an interesting idea, especially if it allows the introduction of new clipping methods improving the convergence of the algorithms.

### ***Concerns and remarks to authors.***

My main concerns with this work are the technical quality and relevance of the contributions. I detail below the most important ones I have.

**Relevance of studying GD instead of SGD.** At first glance, it seems that the authors are studying the well-known DP-SGD procedures and trying to improve the existing knowledge of this widely used algorithm. In fact, this is the method that is presented in Algorithm 1 and that is being used in most of the experiments (Sections 6.1, 6.2, 6.3). However, the theoretical contribution only studies full gradient descent, which is much less relevant in the context of deep learning because it is never used in practice due to its computational complexity. As a result, I think the authors overstate their contribution by stating that they provide a "convergence theory for deep learning DP". I think their analysis is more applicable to simple classification or regression tasks with small datasets.

**Regarding the NTK framework.** The idea of studying the convergence dynamics of differentially private deep learning through the lens of gradient flow seems like a good idea, but I am not sure the authors properly used and analyzed existing tools from control theory. Here are some of my comments below.

1. To build their whole theoretical framework, the authors explain that the derivative of the loss function is characterized by $\dot{L} = \frac{\partial L}{\partial f} HC \frac{\partial L}{\partial f}^{T}$. Then, they introduce a set of new concepts extending the notion of matrix positivity to non-symmetric matrices (Definition 4.3), to analyze HC (because it might not be symmetric). However, from my understanding of the literature, it is not relevant to study HC directly. In fact, we can always rewrite the derivative of the loss function as $\dot{L} = 1/2 \frac{\partial L}{\partial f} (HC + (HC)^T) \frac{\partial L}{\partial f}^{T}$. Therefore, the analysis of $(HC + (HC)^T)$ is sufficient to characterize $\dot{L}$ and does not require the introduction of new concepts since $(HC + (HC)^T)$ is indeed symmetric.
2. It is unclear to me how certainty equivalence can be applied to provide a proof for Fact 4.1. I think that certainty equivalence is related to the fact that the optimal control is the same; that is, the system with zero-mean additive noise has the same optimal control law as the system without noise. But, as I understand it, equivalence of optimal control laws does not mean equivalence of dynamics. Moreover, certainty equivalence does not hold true when the dynamical system is nonlinear, which is indeed the case here.

**Unfair comparison between global and local clipping.** The paper presents a new method of clipping called global clipping and compares it to local clipping (previous method) both theoretically and empirically.  I have some concerns on both sides, explained below.

1. From a theoretical point of view, the authors claim that global clipping offers better convergence guarantees than local clipping. To justify this claim, they present two theorems stating that local clipping can have a bad impact on convergence (Theorem 1) while global clipping does not (Theorem 2). However, the comparison does not seem fair to me. Indeed, in order to show that global clipping does not have a bad impact on convergence, the authors make the additional assumption that $Z$ (one of the clipping parameters) is large enough so that no vector is dropped during the procedure.  In fact, this assumption is equivalent to saying that there exists a real number $M < \infty$ such that for all $i$ and $t$, $\Vert v_t^{(i)} \Vert \leq M$. In this particular case, taking $Z \geq M$, the global clipping amounts to rescaling each gradient with a fixed constant; therefore, it does not hinder the convergence (it is actually equivalent to changing the learning rate). As the authors point out in Footnote 5, this assumption is central to the proof of Theorem 2 and cannot be relaxed. But if we were to assume the existence of such an upper bound $M$ on the gradients in Theorem 1 as well, then we could also show that local clipping does not impede the learning procedure simply by taking $R \geq M$. As a result, the paper does not present a fair comparison of the two clipping methods, and as I understand the results at this point, there is no way to argue that global clipping is better than local clipping from a theoretical viewpoint.
2. From an empirical standpoint, while I agree that global clipping seems to help in classifier calibration, I do not think it can be considered to have similar performance as local clipping. Looking at the experimental results (e.g., Figures 3 and 5), it appears that DP-SGD with global clipping generally takes longer to converge than DP-SGD with local clipping. While this may not be very important in standard deep learning, it is critical when considering privacy because each learning step increases the final privacy budget. Therefore, DP-SGD with global clipping will use a larger privacy budget to achieve the same level of accuracy as DP-SGD with local clipping.




**Summary Of The Paper:**

This paper studies the problem of understanding the convergence of differentially private deep learning models. The most common technique to ensure differential privacy (DP) in deep learning is to adapt the SGD algorithm by clipping the gradient at each learning step before perturbing it with Gaussian noise. The present work proposes to study the impact the clipping step and the noise injection can have on the convergence of the algorithm. In doing so, the authors claim to make several contributions summarized below.

1. Provide a characterization of the training dynamics of differentially private gradient descent methods through the lens of gradient flow theory, using neural tangent matrices (NTK).  The authors claim that this framework enables a fine-grained analysis of the impact of clipping and noise injection on the convergence of deep learning algorithms.
2. Develop a new clipping method called "global clipping" that improves convergence and mitigates the lack of calibration of differential private gradient descent methods. Moreover, this new approach does not change the DP guarantees of the gradient descent algorithm, is easy to implement and is as computationally efficient as the previous clipping methods (called "local clipping" in the paper).


**Summary Of The Review:**


Overall, while I think this paper explores an interesting question, I also think it overstates its contributions and impact. In particular, I think that the utility of global clipping is not sufficiently well supported by theory or experiments. In addition, I think that the use of gradient flow to analyze the dynamics of DP-SGD is interesting but lacks clarity at this time. For the above reasons, I will argue for rejection.

---

> ### Author Response · Authors · 2021-11-13
> **Response to Reviewer b1Rw -- Part 1**
>
> **Relevance of studying GD instead of SGD**
>
> Thank you for this constructive comment! We agree that the theoretical contribution is on the GD only, but the insight we gained here is applicable to SGD and other optimizers in general. To be specific, it is well-known (see [1,2,3,4]) that SGD’s gradient flow follows a stochastic differential equation, that is similar to GD’s ordinary differential equation: $dw_t=-\frac{\partial L}{\partial w_t}dt+\sqrt{\frac{\eta}{n}}\Sigma(w_t)^{1/2}dW_t$, where $\Sigma$ is covariance matrix and $W_t$ is the standard Brownian motion. This means that, in DP training, the SGD loss dynamics is $\mathbb{E}\dot{L}_t=-\frac{\partial L}{\partial f}HC\frac{\partial L}{\partial f}^\top$. Therefore, our framework and discussion about the effect of clipping, i.e. the NTK matrix changes from $H$ to $HC$, are all valid under SGD. The analysis of GD is just relatively more straightforward. We will discuss this important extension to the main text with a highlighting connection to the experiments, where DP-SGD not GD is already used.
>
> [1] Mandt, Stephan, Matthew D. Hoffman, and David M. Blei. "Stochastic gradient descent as approximate bayesian inference." arXiv preprint arXiv:1704.04289 (2017).
>
> [2] Smith, Samuel L., and Quoc V. Le. "A Bayesian Perspective on Generalization and Stochastic Gradient Descent." In International Conference on Learning Representations. 2018.
>
> [3] Chaudhari, Pratik, and Stefano Soatto. "Stochastic gradient descent performs variational inference, converges to limit cycles for deep networks." In International Conference on Learning Representations. 2018.
>
> [4] Xie, Zeke, Issei Sato, and Masashi Sugiyama. "A Diffusion Theory For Deep Learning Dynamics: Stochastic Gradient Descent Exponentially Favors Flat Minima." In International Conference on Learning Representations. 2020.
>
> **Regarding the NTK framework.**
>
> 1: We thank the reviewer for pointing out that the $HC$ can be made symmetric (though only in the quadratic form scenario). We were aware of this simpler form but did not use it for two reasons. First of all, when deriving Statement 4 in Theorem 1 and Statement 2 in Theorem 2, i.e. whether the terminal training loss is zero, we need a ‘prediction dynamics’ (moved to page 16 in appendix B due to page limit), which takes the form of $\dot{f} = -HC \frac{\partial L}{\partial f}$. This implies $\frac{\partial L}{\partial f}=0=y-f$, and thus zero training loss when the dynamics is stationary, i.e. $\dot{f}=0$. Note that using $HC + (HC)^\top$ is not relevant to this analysis since $-HC \frac{\partial L}{\partial f}\neq -(HC+(HC)^\top) \frac{\partial L}{\partial f}$. Secondly, the symmetry of NTK itself is not essential but more about the mathematical form. It is straightforward to show that the symmetrization $HC+(HC)^\top$ is positive semi-definite if and only if $HC$ is positive in quadratic form. Thus, we choose to use the non-symmetric form of $HC$ for a general and unified analysis.
> In short, the non-symmetric form $HC$ is essential to both the proofs of positivity in eigenvalue and positivity in quadratic form. In contrast, $HC+(HC)^\top$ is only a transformation in form that can be only used for the positivity in quadratic form, even though symmetry is preserved.
>
> 2: We would like to assure the reviewer that we only use certainty equivalence as a concept for analogy but did not use it technically as any kind of **proof** of Fact 4.1 (whose rigorous proof can be found in Appendix B page 15). We definitely understand that the certainty equivalence is a phenomenon observed in the control community, and never indicate there is a “control” in the DP training dynamics.

---

> > ### Author Response · Authors · 2021-12-05
> > **Re: Response to the reviewer b1Rw**
> >
> > We thank the reviewer for the comment again and hope that you find our detailed response useful. We are happy to address any other concerns you have. In the meantime, if all of your concerns are cleared, we sincerely hope that you could reconsider the score. We want to re-emphasize the great novelty in our work -- we give the first NTK-based framework for analyzing DP optimizers, the new global clipping (which is completely different from the existing clipping and makes DP-SGD equivalent to the stochastic gradient Langevin dynamics; note that most existing theoretical works on DP-SGD has to get around the per-sample clipping by assuming unrealistic Lipschitz-ness and convexity conditions), and the significant improvement on calibration error of our method. The minor points, like the small accuracy gap (say, 1% accuracy drop on BERT) should not affect our significance. We really appreciate your cautionary evaluation of our work, and believe you will understand the significance here.

---

> ### Author Response · Authors · 2021-11-13
> **Response to the reviewer b1Rw -- Part 2**
>
> **Unfair comparison between global and local clipping**:
>
> We thank the reviewer for the insightful comment.
> 1. For the ease of discussion, let’s assume $Z=M=\sup_{i, t}\|v_t^{(i)}\|$. We agree that in this case, if $R\geq Z$, then local clipping does not impede convergence as $C_i=1$. In particular, when $R=Z=M$, then local and global clippings are the same. However, in practice, we have $R<Z$. This is because the learning rate is not infinitesimal and thus the noise is not ignored -- note that the noise magnitude is $\sigma R$ in Equation (4.1) and a small $R$ is preferred (see arxiv: 2110.05679, Figure 4). We acknowledge that though this special case when $R\ge M$ does not affect our theoretical claims, it is still worth mentioning in our empirical claims and discussion. We will add a remark in our updated paper.
>
> 2. We do believe global clipping “can be considered to have **similar** performance as local clipping” in the sense that the accuracy difference is small: e.g. 1% on MNIST; 1% on SNLI. Although it may take longer training (and privacy loss) to match the **same** accuracy, we would like to point out two key aspects to consider. First, accuracy is not the only performance metric to take into account, that’s why we consider privacy, and, likewise, calibration in the first place, especially given that the accuracy difference is very small. Besides, DP-SGD(GD) with our global clipping is much more amenable for theoretical analysis, as it can be viewed as a regular SGD with additive Gaussian noise, like stochastic gradient Langevin dynamics (SGLD) in Bayesian neural networks. We hope the reviewer can appreciate this generality in its theoretical form and flexibility in its usage. Our global clipping also builds the bridge between DP-SGD and SGLD and opens doors to new analysis on DP optimizers.

---

### Official Review · Reviewer_Aa4t · 2021-11-02

**Correctness:** 3
**Technical Novelty And Significance:** 3
**Empirical Novelty And Significance:** 4
**Recommendation:** 5
**Confidence:** 3

**Main Review:**

Strengths:
  - The proposed global clipping is very simple and intuitive to implement.
  - The (lack-of) effects of Gaussian noise on the gradient flow is a very neat fact.
  - The analysis of the proposed global clipping is analysed in various settings (with even more details in the appendix).
  - Experiments are very promising.

Weaknesses:
  - Claims on convergence rely heavily on the NTK matrix being positive-definite. However, is this the case in practice? I know that the Fisher information matrix has mostly close to zero eigenvalues (e.g. "Universal Statistics of Fisher Information in
Deep Neural Networks: Mean Field Approach 2019"), but are there similar characterizations of the spectrum for NTK?
  - Similarly to the above, does the positive-definiteness hold in practice?
  - Theorem 2 relies on the global maximum norm bound $ Z $ for the batch gradients. How often does this happen in practice?
  - The claim of Theorem 3 seems to be worded quite strongly. My initial interpretation of it was that local clipping is $(\varepsilon, \delta)$-DP iff global clipping is $(\varepsilon, \delta)$-DP. However, I don't believe this is the case given the proof.

Other / Minor:
  - Inconsistent use of oxford comma (abstract + intro versus rest of paper).
  - For Figure 1, it would help to emphasis the global clipping by making the grey circle smaller.
  - Algorithm 2 mentioned in main text needs a pointer to appendix.
  - It would be worth stating that you are looking at fair loss function where no loss is incurred for perfect prediction (e.g. "Information, Divergence and Risk for Binary Experiments 2011"). Especially so since this is key to the proof of Theorem 1 statement 4.
  - A few typos in the appendix:
       - Proof of Theorem 2 statement 1: at $ x^{T} x \mathbf{H} x $;
       - "citep" above Lemma B.1;
       - Lemma B.1 "supreme" -> "supremum".
  - Top of Section 4 "yet and our" -> "yet our".

**Summary Of The Paper:**

The paper analysis differentially private (DP) optimization convergence in neural networks by examining the neural tangent kernel (NTK). In particular, the papers analysis focuses on DP optimization which utilize gradient clipping and the Gaussian mechanism for gradient updates. Through the analysis of the NTK, an alternative clipping operation is proposed which has guarantees for convergence of the loss function. In addition, this alternative clipping operation provides a DP guarantee. Experiments in the paper then verify that the proposed clipping performs well in practice.

**Summary Of The Review:**

Overall I would recommend an accept. The simplicity of the approach and the improvements shown in the experiments show that the global clipping has promise. However, there are a couple of aspects in the paper which need to be addressed. In particular, I believe that the text of Theorem 3 is worded too strongly.

---

Edit: I have changed my score from 6 -> 5 as a result of reviewer discussion. In particular, due to how the assumptions of Theorems 1 & 2 are not the same due to the assumption on the gradient norm bounds. Further discussion / analysis is needed for the violation or removal of this particular assumption (following the critics presented in LL2P and b1Rw's reviews).

---

> ### Author Response · Authors · 2021-11-13
> **Response to Reviewer Aa4t**
>
> We thank the reviewer for these practical questions. We will address all minor issues in the camera-ready version as well.
>
> **Weakness 1:** We would like to refer to a long line of NTK papers that show the positive definiteness of NTK matrix: on fully-connected layers [1,2,3,5], on CNN [2,3], on RNN [4], on SGD [3,6], on different losses [3], and so on. The idea is that with wide-enough network and small learning rate, the NTK matrix is guaranteed to be positive definite. We would like to highlight that in the regular training, by the definition of NTK as an inner product, even if it is not strictly positive definite, at least it must be positive semi-definite. However, the existing local clipping cannot even guarantee to preserve this positive semi-definiteness, which may lead to, for example, the non-monotone convergence as shown in Figure 5,7, and 9.
>
> **Weakness 2:** In practice, for GD and SGD (with sufficiently small learning rate), the NTK matrix is likely to be positive definite, as indicated by the monotonically decreasing loss in many experiments, also reflected in our experiments.
>
> **Weakness 3:** In our experiments, it is not difficult to select $Z$ that has near-optimal performance. For example, on MNIST setting $Z=250$, is enough for finite epochs and results in a peak accuracy. In practice, we only need $Z$ that approximately upper bounds most of the gradient norms. To give more details, on MNIST we run 60 epochs (more than 10,000 iterations). Even if at some iteration $Z=250$ is still below the maximum gradient norm, we find $Z$ is above the maximum gradient norm for 95% of the iterations. Our analysis is at least a close approximation to the actual dynamics when $Z$ is large enough.
>
> **Weakness 4:** We thank the reviewer for this clarification comment. Theorem 3 is surely correct and we will polish the wording to make the statement easier to understand without confusion for the general audience. In fact, our theorem claims that, taking one step of DP-SGD with local clipping incurs the **exact** same amount of privacy risk as one step of DP-SGD with global clipping. Therefore, we can assert that the entire DP training with either local clipping or global clipping for the same number of iterations will result in the exact same DP guarantee. Please let us know if this has resolved your concerns.
>
> [1] Du, Simon S., Xiyu Zhai, Barnabas Poczos, and Aarti Singh. "Gradient Descent Provably Optimizes Over-parameterized Neural Networks." In International Conference on Learning Representations. 2018.
>
> [2] Du, Simon, Jason Lee, Haochuan Li, Liwei Wang, and Xiyu Zhai. "Gradient descent finds global minima of deep neural networks." In International Conference on Machine Learning, pp. 1675-1685. PMLR, 2019.
>
> [3] Allen-Zhu, Zeyuan, Yuanzhi Li, and Zhao Song. "A convergence theory for deep learning via over-parameterization." In International Conference on Machine Learning, pp. 242-252. PMLR, 2019.
>
> [4] Allen-Zhu, Zeyuan, Yuanzhi Li, and Zhao Song. "On the convergence rate of training recurrent neural networks." arXiv preprint arXiv:1810.12065 (2018).
>
> [5] Chen, Zixiang, Yuan Cao, Quanquan Gu, and Tong Zhang. "A generalized neural tangent kernel analysis for two-layer neural networks." arXiv preprint arXiv:2002.04026 (2020).
>
> [6] Cao, Yuan, and Quanquan Gu. "Generalization bounds of stochastic gradient descent for wide and deep neural networks." Advances in Neural Information Processing Systems 32 (2019): 10836-10846.

---

> > ### Comment · Reviewer_Aa4t · 2021-11-17
> > **Re: Response to Reviewer Aa4t**
> >
> > Thank you for your detailed response. Just wanted to answer in advanced that my concerns for "Weakness 4" have been addressed.

---

> > > ### Author Response · Authors · 2021-11-22
> > > **Re: Re: Response to Reviewer Aa4t**
> > >
> > > Thank you for the response! We are glad to know that one of your concerns is addressed. Please feel free to let us know if there is any other outstanding concern. We would like to elaborate more and resolve all your questions.

---

> > > > ### Author Response · Authors · 2021-12-05
> > > > **Re: Re: Response to Reviewer Aa4t**
> > > >
> > > > We thank the reviewer for the comment again and hope that you find our detailed response useful. We are happy to address any other concerns you have. In the meantime, if all of your concerns are cleared, we sincerely hope that you could reconsider the score. We want to re-emphasize the great novelty in our work -- we give the first NTK-based framework for analyzing DP optimizers, the new global clipping (which is completely different from the existing clipping and makes DP-SGD equivalent to the stochastic gradient Langevin dynamics; note that most existing theoretical works on DP-SGD has to get around the per-sample clipping by assuming unrealistic Lipschitz-ness and convexity conditions), and the significant improvement on calibration error of our method. The minor points, like the small accuracy gap (say, 1% accuracy drop on BERT) should not affect our significance. We really appreciate your cautionary evaluation of our work, and believe you will understand the significance here.

---

### Official Review · Reviewer_LL2P · 2021-11-05

**Correctness:** 4
**Technical Novelty And Significance:** 2
**Empirical Novelty And Significance:** 2
**Recommendation:** 3
**Confidence:** 3

**Main Review:**

I think that using the NTK framework to analyze the converge of DP-SGD is a very interesting direction. However, the theoretical claims seem not surprising and I question how useful the global clipping strategy is for practical purposes. First, according to theorem 2, the convergence of SGD with global clipping is only guaranteed when we can find a parameter Z big enough such that all sample gradients have norm less than Z. However, it is not possible to choose such a Z in advance before running the algorithm, therefore theorem 2 it’s not very informative for the practical algorithm (i.e., what should we set Z too?). In practice, Z is a hyperparamter that needs to be tune before running the algorithm since, according to figure 3 (rightmost plot), the choice of Z has a big influence in the final model accuracy. By using the original local clipping strategy, one does not need to worry about finding an optimal Z, thus one could claim local clipping is more practical.

On the empirical evaluation results are also unsatisfactory in my view. Somehow, using global clipping results in lower test loss that using local clipping but local clipping seems to give better accuracy, this phenomenon is not explained in the paper. Specially in the hard CIFAR10 setting, using global clipping does not seem to make a big difference. Furthermore, running global clipping requires knowledge of the Z parameter but that’s not required for local clipping. Therefore, I’m not convince that I should use global clipping vs local clipping for practical applications.



Questions:
- The paper makes then case that global clipping reduces the clipping bias that occurs with local clipping. However when Z is small, removing samples from the batch can also creating a bias. How can we compare then bias of both methods and can one create an example where global clipping would change the gradient direction more than (or add more bias) local clipping would?

- I don’t understand what figure 1 is trying to convey. My understanding is that the figure is comparing the average clipped gradient with global and local clipping. However it would be good to compare against some baseline. For example, we could compare against  the gradient of the average gradient without clipping.





Other comments:
I was confused by definition 4.3. Is the definition 4.3 equivalent to positive-semi-definite matrix?

**Summary Of The Paper:**

The paper analyzes convergence of DP-SGD using the NTK framework. The general idea is that the learning dynamics of SGD converge when the NTK matrix is positive-semi-definite.  They  characterize the effect of clipping and noise and Propose global clipping strategy. The global clipping idea consists of eliminating samples that have gradient norm bigger than some threshold and then applies a common clipping multiplier for each sample gradient. The authors claim that global clipping reduces gradient bias.


Gradient Clipping Summary: DP-SGD consist of the following steps: Over T rounds: 1) the algorithm samples a batch of samples from a dataset I_t. 2) it computes the gradient of each individual samples. 3) Each gradient sample is multiplied by a number such that the final gradient has norm bounded by R. 4) Finally, all clipped gradients are aggregated, noise is added to the average clipped gradient and the result is applied to update the model.

Difference between local and global clipping:  In local clipping, the gradient of each sample is multiplied by a factor specific to that sample such that the gradient  norm  of each sample is bounded by R. In global clipping, there is an additional parameter Z, each sample in a batch with gradient norm smaller than Z are multiplied by a constant factor R/Z and samples with gradient norm bigger than Z are not used. Therefore in global clipping each sample in a batch has gradient norm bounded by R. The intuition behind global clipping is that each gradient in a batch gets multiplied by a constant factor, this means that the direction of the aggregated gradient should be preserve.


Convergence Analysis: The convergence analysis uses the Neural Tangent Kernel (NTK) framework. The premise is that, the convergence of SGD is controlled by the NTK matrix in that the condition for convergence is that the NTK matrix is positive-semi-definite. The paper provides two main theorems: The first one claims that local clipping cannot guarantee convergence because local clipping can break the conditions for convergence under the NTK framework. The second theorem says that when the parameter Z is large then global clipping can guarantee convergence because each gradient is multiplied by a constant factor and does not affect the positive-semi-definitess of the NTK matrix.

Experiments: The paper empirically evaluates global clipping on the MNIST and CIFAR10 dataset. They


**Summary Of The Review:**

The paper has a very interesting theme but I find the analysis and conclusion unsatisfactory. Furthermore, the empirical evaluation is not more convincing.

---

> ### Author Response · Authors · 2021-11-13
> **Response to Reviewer LL2P**
>
> We would like to thank the reviewer for the questions.
>
> **Tuning of $Z$:**
> The reviewer is correct that the newborn global clipping at its current form does need an additional hyper-parameter $Z$. Nevertheless, we argue there are two reasons that introducing $Z$ is not too troublesome. First of all, the accuracy is not very sensitive to the choice of $Z$. In our Figure 3, any $Z$ between 150 and 250 results in high accuracy with negligible difference. Therefore tuning $Z$ for a good accuracy is easy, especially when compared to the tuning of the existing hyper-parameters like $R$ or $\sigma$ in local clipping. Secondly, there has been a line of work to determine the clipping norm $R$ adaptively and privately, without manually tuning (see, e.g. arxiv: 1905.03871). This approach can be easily adopted into choosing $Z$ automatically without tuning as well, we may consider this extension in our future works.
>
> **Empirical evaluation results:**
> We thank the reviewer for pointing out this interesting phenomenon. In fact, even in non-private deep learning, it is not uncommon to have good accuracy despite high loss, and it is not well-understood. We actually explained part of it in our discussion of calibration: suppose we have two models A and B, both give similar accuracy, but A has higher cross-entropy loss than B. It is often that B is fairly accurate on all samples, resulting a low $-\log \hat{p}_i$ for all $i$, but A may have very bad prediction on some samples (e.g. the true class is 7 but only 0.01 probability is assigned in prediction), resulting in high $-\log \hat{p}_i$ (e.g. -log(0.01)) for $i$ that boosts up the loss. This is also reflected in the much worse calibration of model A than model B. We believe this is the case for local clipping (similar to model A), and global clipping (similar to model B). We want to highlight that accuracy is not the only performance metric to take into account by this work, that’s why we consider privacy, and, likewise, calibration in the first place, especially given that the accuracy difference is very small. This is one of the key contributions of our global clipping.
>
> **Question 1:**
> We thank the reviewer for this interesting observation. In practice we tend to choose a large $Z$. But the trade-off of the bias from local clipping and bias of screening from $Z$ is indeed also interesting. Empirically, the bias introduced by local clipping can be severe, e.g. breaking monotone convergence or even diverging in Figure 5,7, and 9. In contrast, the bias introduced by global clipping is smaller: choosing $Z$ from a range of values that do not necessarily upper bound the gradient norms strictly can still result in satisfying accuracy and low loss (see Figure 3).
>
> **Question 2:**
> We thank the reviewer for the suggestion. Figure 1 serves as a step-by-step visualization that our clipping preserves direction of averaged gradient before and after **clipping** while the local clipping does not. It also helps the readers to understand Algorithm 1. It is indeed due to the page limit that we didn’t put on the comparison with averaged gradients without clipping. We will include this in the camera ready version.
>
> **Clarification on Definition 4.3:**
> Definition 4.3 is the same as positive semi-definiteness only when the matrix is **symmetric**. However, DP deep learning with clipping inevitably needs to deal with a non-symmetric NTK matrix $HC$ (as a consequence of local clipping), and it cannot be made to be symmetric for the $f$’s dynamics (please see page 16 in Appendix B and also [our comment to another reviewer](https://openreview.net/forum?id=2s4sNT11IcH&noteId=UjdHCqeCVoJ)). Therefore we proposed the two different notions of positivity (positivity in eigenvalues, and positivity in quadratic forms) to study two different convergence behaviors in our theorems (e.g. Statement 3 and 4 separately in Theorem 1).

---

> > ### Author Response · Authors · 2021-12-05
> > **Re: Response to Reviewer LL2P**
> >
> > We thank the reviewer for the comment again and hope that you find our detailed response useful. We are happy to address any other concerns you have. In the meantime, if all of your concerns are cleared, we sincerely hope that you could reconsider the score. We want to re-emphasize the great novelty in our work -- we give the first NTK-based framework for analyzing DP optimizers, the new global clipping (which is completely different from the existing clipping and makes DP-SGD equivalent to the stochastic gradient Langevin dynamics; note that most existing theoretical works on DP-SGD has to get around the per-sample clipping by assuming unrealistic Lipschitz-ness and convexity conditions), and the significant improvement on calibration error of our method. The minor points, like the small accuracy gap (say, 1% accuracy drop on BERT) should not affect our significance. We really appreciate your cautionary evaluation of our work, and believe you will understand the significance here.

---

### Decision · Program_Chairs · 2022-01-20

**Decision:**

Reject

**Comment:**

The major concern with this paper is the unfair comparison between global and local clipping (at least from the theoretical point of view). The assumption that the norms of the gradients are bounded in Theorem 2 is too strict for the following reasons. Clipping has been introduced exactly because we cannot assume the norm of the gradient to be bounded by a fixed constant  in the first place. Accordingly, comparing two clipping methods under the bounded gradient assumption does not seem relevant. Further, the two methods are not studied under the same set of assumptions (In Theorem 1, the norm of the gradient is not assumed to be bounded, but in Theorem 2 it is).
A fair comparison needs to be presented to make the case for the proposed method.